# Identification of Hybrid Polymer Material STERED and Basic Material Properties Used in Road Substructures or Pavements

**DOI:** 10.3390/polym16050663

**Published:** 2024-02-29

**Authors:** Daniel Papán, Martin Decký, Daniel Ďugel, Filip Durčák

**Affiliations:** 1Department of Structural Mechanics and Applied Mathematics, University of Zilina, Univerzitna 8215/1, 010 26 Zilina, Slovakia; daniel.dugel@uniza.sk (D.Ď.);; 2Department of Highway and Environmental Engineering, University of Zilina, Univerzitna 8215/1, 010 26 Zilina, Slovakia; martin.decky@uniza.sk

**Keywords:** hybrid polymer, mechanical properties, static load test, stress–strain diagram, modulus of elasticity, compressive strength

## Abstract

Modern road construction uses a large number of polymer-based materials. Material composition depends on their roles. Among the most important functions of road body materials is to transfer all loads safely to the subgrade. A thorough understanding of material properties in various climates is crucial for this purpose. In the automotive industry, polymer residues from recycling can be used to make innovative materials, such as STERED, a hybrid polymer composite. Drawing on the porous nature of this material, this paper investigates its mechanical behavior. For road construction, the compressive properties of the material are most important. The paper presents the results of a detailed analysis and experimental research of the STERED material from in-lab tests. Successful research will lead to the inclusion of the material in road body compositions with excellent retention properties, vibration damping, and potential in circular economy.

## 1. Introduction

Within a holistic view of structural roadway design, the authors started from the premise of using traffic surfaces to increase the climate adaptability of transportation infrastructure. The changing climatic conditions of Central Europe, a decrease in total rainfall, and a significant increase in torrential rainfall are facts that need to have already been taken into consideration at the road design stage [1]. In the research, the authors focused on the verification of the mechanical efficiency of roadway structures using an innovative material called STERED (PROTERED). The STERED material is a type of polymeric foam. It is a recycled product from the automotive industry, which is produced in the manufacture of new automobiles and also in the recycling of old vehicles. This process produces large quantities of textile materials of different qualities. Originally, these materials were used in cars to insulate sound, vibrations, and heat, and they met strict requirements concerning health and resistance to external influences. The STERED material is currently used in sound-absorbing wall cladding, as a railway rolling noise absorber, in noise barriers, or as thermal insulation in the construction of partition or attic systems. Owing to, among others, the STERED technology, Slovakia is one of the countries that are able to meet the Directive 2000/53/EC of the European Parliament and of the Council on the disposal of old vehicles [2].

To emphasize the benefits of using the STERED material in road construction, it is necessary to consider the significance of the material in roadway engineering and its sustainable and environmental properties. Global development in the construction industry, including the construction of new roads and the rehabilitation of elements of integrated transportation infrastructure, requires innovative approaches in the field of material engineering of transportation structures [3]. The vast majority of recent innovation studies have focused on technological innovations in manufacturing [4] without a broader consideration of the holistic view of sustainability in the construction industry with an emphasis on the circular economy. As the planet is experiencing an increasing scarcity of construction resources and a surplus of waste, the concept of ‘circular economy’ (CE) has emerged from growing concerns about the efficiency of resource usage, waste management, and material security of regions [5,6,7,8]. Kibert [9] laid the foundations of sustainable building practices based on the reuse of resources (material and energy), the use of renewable and recyclable resources, and the minimization of the carbon footprint. Vanegas and Pearce [10] presented an SC based on resource depletion and degradation that impact the built environment and human health.

In the field of sustainability of material and environmental engineering of transportation structures, the authors address the possibilities of using industrial waste (waste from automotive production) in roadway structures and noise barriers [11,12]. In the course of the production of automobiles and their textile parts, as well as in the recycling of vehicles at the end of their service life, which involves separated textile parts, a variety of textile materials with different specific qualities are produced. After their recycling, these industrial wastes can be successfully used as effective climate-active road construction components. In the conditions of Slovakia, STERED (abbreviation in Slovak: S—spracovanie, TE—textilného, RE—recyklátu, D—do dosiek; PROTERED—processing, textile, recycled, into boards, next in text using the original Slovak abbreviation) can be considered as an innovative climate-active material. Textiles used in the automotive industry have to meet high standards in terms of sound and thermal insulation and resistance to mold and mechanical damage. STERED significantly reduces vibrations and is also hygienic. In the everyday use of these materials in the automotive industry, we already take these properties for granted, but a lesser-known fact is that these materials are characterized as difficult-to-fibrillate hardened textiles and combined textile-non-textile components, whose further standard textile processing is impossible despite their high quality.

The average weight of the fabrics in one car is around 25 kg, and the production process of a new part generates an average of 2–3 kg of technological waste. As this material is not reused, it either ends up in landfills or is recovered for energy in special incinerators. One solution to this problem is the STERED^®^ complex process line, which produces innovative, climate-adaptive composite materials by bonding textile fabrics with polyurethane binders [13].

In order to make a credible assessment of the structural roadway design, it is necessary to know the elastic moduli concerning the expected levels of traffic loading induced by the passing of vehicles. In order to make the most of the retention and evaporative properties of the PROTERED (STERED) material, the initial intention was to place it as close to the surface of the road as possible. Based on an indicative quantification of elastic moduli of 5 to 10 MPa, it was found that PROTERED (STERED) should be placed in unbound road subbases [14]. The paper is focused on the analysis of the mechanical properties of polymeric materials and on the compression tests of STERED, which will be subjected to freezing and thawing in the experiment.

### 1.1. Polymer Foam Structure

The use of polymeric foams in engineering applications is steadily increasing, mainly due to their energy-absorbing ability, low weight, and relatively low production costs. The main applications of the soft and semi-rigid foams include insulation, packaging, and cushioning, while rigid foams are also used in structural applications as well as the core of lightweight sandwich structures [15]. The choice of material depends on the application and should be based on factors such as operating force and pressure, duration and direction of loading, design constraints, environmental conditions, and costs. Therefore, a thorough understanding and evaluation of the mechanical behavior under different loading conditions is required in order to determine the range of applications in which certain materials would be the most suitable [16]. Polymeric foam can consist of either closed or open cells, as shown in Figure 1. The open-cell foam structure creates continuous channels where air can flow freely in the cell network [17]. Closed porous materials are mainly used as thermal or acoustic insulation. The open structure is permeated with moisture and vapor and has low water absorption [18]. According to their mechanical strength, foams are divided into flexible or rigid [19].

### 1.2. Behavior of Polymer Foams under Compression

Polymeric foam materials may exhibit local variability, which occurs due to different manufacturing methods and different structures of the material, resulting in different mechanical properties [20]. Polymeric foams have a wide range of macro-structures, such as open-cell, partially open-cell, or closed-cell structures.

The typical strain and stress progression of a polymeric foam subjected to a compressive load is shown in Figure 2. This curve can be divided into three areas:Linear elastic—Usually small deformations (5–10%). In this elastic phase, the slope of the stress–strain curve is characterized by Young’s modulus of the foam elasticity.Plateau-collapse—In this area, large plastic or elastic strains occur with slightly increasing stress. This phase absorbs much of the energy, and the cell walls begin to collapse. This area can be defined as the area in which the stress does not increase significantly with increasing strain [21].Densification—the structure reaches a section of compaction where the bonds begin to bump into each other. In this way, stress is transferred to all cellular structures (not just the cell walls), resulting in a dramatic increase in strength and, ultimately, damage to the material. At this last stage, the foam begins to exhibit a modulus of elasticity approaching that of the solid material from which it is made [22].

The scheme of the stress–strain curve of uniaxial compression of an open-cell elastic foam (Figure 2), which illustrates these three areas, is shown in Figure 3. Current theories about the shape of the strain curve of open-cell foam under loading are based on the distinction between bending and buckling. Previous research has shown that the initial linear elastic portion of the stress–strain curve is the result of the bending of the cell walls, which are oriented perpendicular to the loading direction, while the flat portion of the stress–strain curve (the breakdown area of the pores) is the result of buckling of the cell walls, which are oriented parallel to the loading direction [23].

A comparison of the images of the 0%, 2%, and 4% strain (Figure 3i–iii) clearly shows that the initial stages of compression are contained by a small number of bends in the cell walls, which are tilted in the direction of compression. Examples of bent walls are shown by the white arrows in Figure 3iii. This corresponds to the linear elastic portion of the macroscopic stress–strain curve. From the comparison of the images of the 10% and 23% strains (Figure 3iv,v), it is clear that the next phase of the deformation process proceeds with a more pronounced bending of the longer cell walls. The paper states that the images presented here demonstrate the existence of a collapse zone. A detailed comparison of the images of the 10% strain (Figure 3iv) and the 23% strain (Figure 3v) shows that an entire band of cells has undergone severe deformation, while other parts of the structure are almost unaffected by the increase in the total load. Comparing the images of the 23% and the 40% strain (Figure 3v,vi), the collapse can be seen to spread out throughout the structure. At the 63% strain (Figure 3vii), the cell walls begin to bump into each other, and the deformation is clearly in the densification phase. At 80% deformation (Figure 3ix), the structure is very densely compacted, and it is difficult to distinguish individual cell walls [24].

It is expected that the material to be subjected to experimental analysis will present a similar process of strain-dependent stress. However, various factors, such as different material density, strain rate, and cellular structure, which may affect this assumption, need to be taken into consideration. These factors affect the magnitude of the elastic deformation and the stress at which the magnitude of the plastic transformation is achieved.

### 1.3. Effect Caused by Freezing and Defrosting on Porous Composite Material

Exposure to frost plays an important role in the degradation of porous materials under severe climatic conditions [24]. During freezing and thawing, a repeated process of expansion and contraction occurs, which can cause the spreading of cracks and enlarged pores inside the material. This process can lead to the degradation of the material and deterioration of its mechanical properties, such as strength and crack resistance. In porous polymer composite materials, the effect of freezing and thawing can cause an increased pore volume and decreased density of the material. This can lead to deterioration of its mechanical properties and reduction of its durability. Therefore, it is important to consider the effect of freezing and thawing in the design and use of porous polymeric materials and to take it into account in their testing and performance evaluation.

### 1.4. Determination of Mechanical Stress on the Construction Layer of the Roadway

Currently, the relevant provisions for the assessment of roadway constructions are codified in the Slovak Road Act No. 135/1961 Coll [25]. The design of roadways is carried out according to the valid Slovak technical standards, which are technical regulations and objectively ascertained results of research and development for road infrastructure. The decisive standard STN 73 6114:2021 [26] states that the roadway should be designed to withstand the loads and impacts that can be expected to occur during its use with the required level of reliability. The authors have presented their objectively observed research results contributing to the sustainability of all types of roadways described in [27,28,29,30].

Designing roadways according to the STN 73 6114:2021 standard [26] is based on the traffic significance of the roadway, the traffic load on the roadway, climatic conditions, technological possibilities, the possibilities of using local materials, and the protection of health and the environment. Design parameters and reliability conditions are determined for design situations, which can differ for different types of roadways (asphalt, cement concrete, pavement, etc.), depending on the characteristics of the load and the environment. The output of the roadway structure design calculation is the response of the roadway to the load or the changes in climatic conditions. In terms of reliability, it is used to assess the technical correctness of the roadway structure design. Only design methods that comply with this standard may be used in roadway design.

The mechanical performance of asphalt and paved roadways shall be assessed through the assessment of operational performance and fatigue cracking. The criterion is met when the calculated radial stress *σ_r_* of the cemented layer at the underside of the layer under consideration (i) is less than the tensile flexural strength of the material *R_i_* reduced by the fatigue coefficient *S*. Due to changes in the deformation and strength characteristics of asphalt materials during the year, the roadway needs to be assessed during several seasons. Our specifications for asphalt pavement design consider three characteristic periods—winter (reference temperature *T_ref,z_* = 0 °C), medium conditions (autumn, spring −*T_ref,j_* = 11 °C), and summer (*T_ref,j_* = 27 °C). The roadway is designed correctly if we observe the following:$$\sum_{i=1}^{i=n}q_{j}\frac{\sigma_{r,i,j}}{S_{N,i}R_{i,j}}\leq 1$$
where 

*q_j_*—relative duration of the seasons (*q_z_* = 0.2; *q_s_* = 0.5; *q_L_* = 0.3)

*S_N,i_*—fatigue coefficient of the material, corresponding to the designed traffic load *N_c_* determined from the relation *S* = *a* − *b*.log*N_c_*

*σ_r,i,j_*—tensile stress in the bend at the underside of the cemented layer under consideration (i) determined by calculation [MPa]

*R_j,j_*—design value of the flexural tensile strength of layer (i) for the relevant seasonal conditions (more detail in Section 3) [MPa]

In the roadway, it is mainly to the cover (Figure 4) where the horizontal forces and the tensile component of the radial stresses (Figure 5) are applied; the other layers are practically unaffected by the effect of the horizontal force.

Current valid design and assessment methods for asphalt and pavement roadways consider only truck-induced vertical forces.

## 2. Methodology

The aim of the experiment was to observe the degradation of the examined material resulting from freeze–thawing cycles. Porous materials are known to have many good and useful properties, which are described in the theoretical section. For the preparation of the experiment, it was necessary to use several apparatuses in which the material was compressed and frozen. These apparatuses are described in more detail in Section 2.2. The experiment consisted of 50 small-sized samples, which were divided into four groups. The first group consisted of 20 dry samples. The second group consisted of 10 saturated samples. The other two groups of 10 samples comprised samples that had been subjected to ten freeze–thawing cycles before compression.

### 2.1. Mechanisation Used during the Experiment

The experiment used a KD-20-T3.1 freeze–thawing chamber (Figure 6a) in which the material was subjected to freeze–thawing cycles. The KD-20-T3.1 freeze–thawing chamber is an automatic test equipment designed for freezing resistance and surface durability tests. It is capable of developing temperatures from −25 °C to +30 °C, and the dimensions of the test chamber are 1200 × 600 × 400 mm.

The SAUTER TVM 30KN70N test stand (Figure 6b) was used for destructive experimental measurement of compression strain, which is used for the accurate measurement of tensile and compressive forces [31]. Various force-measuring devices can be mounted on the test equipment. The maximum force it can exert is 30 kN. The maximum distance traveled in the device is 210 mm. Compressive forces and strain were measured by this mechanism.

In addition to these instruments, a digital caliper, a balance, and an electric knife were also used to prepare the samples.

### 2.2. Experimental Samples

Small-scale samples (Figure 7b) measuring 50 × 50 mm were used in the experiment. These were cut using an electric knife. Cutting with the electric knife proved to be the most suitable method (Figure 7a), as it allowed the samples to be cut with adequate accuracy without significantly disturbing the structure of the material on the cutting plane.

After cutting, the samples were weighed, and the digital caliper was used to determine their height, width, and length. A selection of these measurements is shown in Table 1—a selection of samples from each group and its dimensions. The height of the samples was determined by the arithmetic mean of the four values from each corner of the samples, as these values considerably varied. For the sake of clarity, the samples were numbered according to the date of measurement. The following table shows selected values from each group.

### 2.3. Preparation of Samples—Exposure to Simulated External Conditions

There were multiple types of samples; some were prepared just as was described in the previous chapter. From these, some were fully submerged underwater for complete saturation, and some were exposed to freeze–thawing cycles (Figure 8), saturated and non-saturated alike. The preparation of these samples shall be described below.

Regular small-scale samples were prepared using an electric knife as described before as the group number 0. From these, other samples were obtained.

Samples fully submerged in water were prepared by placing small-scale samples underwater using regular tap water. They were left in a container filled with water overnight. The temperature of the water was measured so as not to decrease below regular room temperature. Samples prepared as described were observed to be fully saturated and ready for the experimental measurement.

Fully saturated and untreated ones were used for freeze–thawing processes as well. The number of freeze–thawing cycles, as well as the individual temperature values during the cycle, were determined on the basis of the test regulations for freezing and thawing resistance. The number of cycles was set at 10, with one cycle lasting 24 h.

### 2.4. Experimental Measurements

Each sample was placed in the testing machine in such a way that the force was applied to the entire pressing surface, which was ensured by means of a circular steel plate, whose diameter was a few centimeters larger than the dimensions of the test sample. After fixing the sample (Figure 9), it was subjected to a force of up to 3000 Newton.

The measured values were evaluated using the least squares method in the Excel computer program. The result of this analysis is a working diagram of the dependency between relative strain and stress. The elastic moduli that will be presented in the following chapters are evaluated using the least squares method described in the theoretical section as average tangent elastic moduli.

In the next chapter, the results of the measured values of each group will be presented. It is not possible to see the primary elastic linear area on the working diagrams of the individual samples due to the lack of sensitivity of the test set-up and the fact that in a material with such a structure, this area is at very low stresses and is practically unmeasurable. It is a fact that STERED behaves like a material with an open pore structure, but its initial strength is unmeasurable and almost immediately breaks down. Therefore, only the elastic moduli in the area of pore breakdown and in the area of densification will be further evaluated. Figure 10 shows a working diagram of one of the dry samples. The orange and red dashed lines represent the average tangent elastic moduli. All the following groups were evaluated in the same way.

### 2.5. Discarded Samples

Several samples from each group had to be discarded during the evaluation. The inhomogeneity of the material (Figure 11) affected the course of the experiment on these samples. These samples contained clusters of elements that are part of the material and have been distributed in this way in the material since its manufacture. This is one of the properties that arise when recycled substances from cars are used, STERED being one of them. These inhomogeneities cause certain parts of the material to have different compressive strengths. In the experimental measurement, these samples showed an incorrect test procedure. In Figure 8, it is possible to observe the different material compositions on the cutting plane of one sample. In the left part, there is a cluster of harder materials (polypropylene and rigid PUR foam), while the right part of the sample is composed of synthetic fabrics (polyester). This different material composition led to uneven strain and, consequently, to an incorrect finishing of the sample measurement.

The samples behaved naturally like the other samples at small strains, but as the load increased, they began to resist the load at one location more than at another. This meant that the test progress did not meet the uniaxial stress progress parameters. These samples, therefore, did not exhibit relevant stress values in the large strain areas.

## 3. Results

The results of the experimental measurements of both saturated and untreated samples are presented below and evaluated using the least squares method in the Excel computer program.

### 3.1. Evaluation of Untreated Samples

Figure 12a shows the stress waveform of nine dry samples. In the diagram, the orange-colored samples are those samples where a region of pore breakdown occurred at smaller stresses, and subsequent densification occurred at larger strains, compared to the samples shown in blue. For the blue samples, the initial stresses were greater, and material densification occurred at smaller strains. This behavior can again be attributed to the inhomogeneity of the material. The samples shown in blue were, for the most part, made up of harder materials with higher density. Thus, greater force was required to collapse their pores, and it occurred at smaller strains than those of the orange samples. A more detailed view of the different stresses at the initial strains is shown in Figure 12b.

Table 2 shows the individual values of the elastic moduli. The deviation between the individual values confirms the fact that the inhomogeneity caused by the different properties of the individual parts that make up STERED makes it impossible to define its strength characteristics clearly. In view of the relatively small variations in the values of the modulus of elasticity in the area of densification presented in these tables, it is clear that the magnitude of the modulus of elasticity in densification is not influenced by the degree of strain at which it occurs.

### 3.2. Evaluation of Saturated Samples

During the experimental measurement of the ten saturated samples, the same problem occurred with sample 7 as with the dry samples. Since STERED has excellent retention properties, the course of the experiment, during which a large amount of water flowed out of the sample, was more complicated. The samples were immersed in water for twelve hours prior to the experimental measurement, which increased the weight of one sample by almost five times. In Table 3, the weights of each saturated sample before immersion in water and before compressing the sample, and the weights after the experiment was completed are shown. On average, 34.37% of water flowed out of the samples during the experiment.

Looking at Figure 13, we note that a heterogeneous pattern of stresses in the saturated samples can be seen again. However, the presence of water did not significantly affect this waveform. The saturated samples showed similar values of elastic modulus as the dry samples. The values are expressed in Table 4—values of Young’s modulus of saturated samples.

The orange curve in Figure 14 shows the stress waveform of one of the untreated samples. When comparing the waveform with the blue curve, which represents the stress waveform of the saturated sample, the waveforms are very similar. The elastic modulus values obtained by linear regression of the measured values are represented by the red, yellow, green, and black waveforms.

### 3.3. Evaluation of Freeze–Thawed Untreated Samples

The samples were subjected to ten freeze–thawing cycles, the course of which is described in the paper. Resistance to climatic loading is an important property that influences the suitability of materials for particular applications. This group consisted of ten samples, and once again, only samples with the correct test procedure were considered. Figure 15 shows the stress waveforms for these samples. The stress waveform for sample 2_2, which is marked in green in the graph, was slightly different compared to all the other samples. For this sample, compaction occurred at smaller strains, even though the sample initially struggled with resisting the loading.

When comparing the working diagram of one of the dry samples that were subjected to freeze–thawing cycles (blue curve in Figure 15) with the dry sample that was not subjected to any influences before the test (orange curve in Figure 16), it can be seen from the graph that the waveforms are almost identical.

The elastic modulus values in Table 5 again do not show significant variations compared to the previous groups.

### 3.4. Evaluation of Saturated and Freeze–Thawed Samples

Samples were immersed in water for twelve hours before being subjected to freeze–thawing cycles. However, during freezing and thawing, some of the water in the freeze–thawing chamber leaked into the container in which the samples were stored, and an average of 76% of the water gradually evaporated from the samples at plus temperatures. Table 6 shows the weights of the individual samples before immersion in water, before placing the sample in the freeze–thawing chamber, and the weight of the samples after removal from the freeze–thawing chamber.

The dependency of stress and strain in compression testing of these samples is shown in Figure 17. The resulting values of the elastic moduli, which are again in the same range as the values for the dry samples, are shown in Table 7.

Figure 18 shows stress dependent on strain, with the blue curve showing a saturated sample that has been subjected to freeze–thawing cycles. The orange curve shows the waveform of one of the dry samples. A very similar pattern can be seen once again, which is confirmed by the calculated elastic moduli represented by the green, black, red, and yellow lines in the graph.

## 4. Discussion

In this chapter, the results of the dry samples will be contrasted with the results in the different groups where STERED was exposed to water and freezing.

Figure 19 shows the individual average elastic moduli in the area of the pore breakdown. The results of the values of the samples exposed to water and frost are comparable to the baseline dry samples. The grey column, which presents the average values of the dry samples subjected to freeze–thawing cycles, shows slightly elevated values, but this was mainly due to the material composition, where it was found after back-examination of the samples that the majority of them were composed of mainly harder materials, which have a higher compressive strength. Both groups of saturated samples (orange and yellow columns) show slightly reduced values, but looking at the specific values of the individual samples in the results, where these values are in a similar range as the dry samples, it cannot be concluded that the exposure of these samples to water and frost had an effect on their compressive strength at small deformations. The variation in the values was due to the several times mentioned inhomogeneous composition of the samples.

When comparing the average modulus of the individual samples in the area of densification, the values in all groups were in the same range. It can be seen in Figure 20 that the saturated and dry samples, which are represented by the yellow and grey columns, respectively, showed lower values on average after being subjected to ten freeze–thawing cycles. However, this was again a consequence of the different material compositions of the different samples.

Based on these results, it can thus be concluded that neither water nor exposure of the material to repeated freezing and thawing has a significant effect on its modulus of elasticity. The inhomogeneity of the material had a significant effect on the course of the whole experiment. A part of each group of samples was excluded due to the incorrect test procedure, where the sample was unevenly deformed. This problem could be partially solved in future experiments by increasing the dimensions of the samples, which would eliminate the effect of the non-uniform material composition of the samples as the load would be distributed over a larger contact area. Freeze–thawing cycles at temperatures between −15 °C and +20 °C did not in any way degrade the properties of any of the materials that STERED is composed of, so it would be appropriate to try exposing the material to larger temperature fluctuations and confront the results with the values measured in this experiment. When further examining the properties of this material, it would be advisable to focus on the effect of the speed at which the material is compressed, as this may be one of the factors that will affect its properties.

## 5. Conclusions

The aim of this work was to determine experimentally the basic mechanical properties of the STERED material. Considering the wide range of applications of the material in industry, civil engineering, or transportation infrastructure, we can certainly assume its further use in various applications in transport construction. In order to determine the suitability of this material for a particular application, it is necessary to know its material characteristics. Several tests have been carried out on the samples under different extreme conditions so that the influence of the presence of water and the effects of frost on the material characteristics of STERED can be determined.

The theoretical part describes the classification of porous materials according to different criteria, with a part of it focusing on polymeric foams, which characterize the mechanical behavior of STERED in more detail. Furthermore, the theoretical part contains basic information about uniaxial strength tests, stresses in simple compression, as well as the influence of freeze–thawing cycles on the degradation of the mechanical properties of the porous materials. The mathematical–statistical method of least squares was used to evaluate the data, the aim of which is to approximate pairs of measured data. This knowledge was the basis for carrying out the experimental part of this work as well as for the evaluation of the results [32,33,34,35].

The experimental part was carried out in a university laboratory on 40 small-size STERED samples, which were divided into groups representing the extreme conditions to which the samples were exposed before the compression testing. The samples were cut using an electric knife, which made it possible to prepare the samples with sufficient precision without significantly disturbing the material structure in the plane of the cut. Dry and saturated samples were subsequently compressed in the experiment. The other two groups consisted of samples subjected to freeze–thawing cycles. Based on the measured values, the working diagrams of the individual samples were evaluated using a mathematical model. The results of the experiment depended not only on the accuracy and precision of its execution but also, to a large extent, on the material composition of the individual samples. The STERED board is produced at the dimension of 1200 × 600 × 50 mm. A board of these dimensions has a number of inhomogeneous components which are distributed in unpredictable clusters. These clusters had an adverse effect on the evaluation of the results for several samples and were, in a way, separated from the other measured samples, as these inhomogeneities led to uneven strain during the course of the test and to subsequent distorted results [36,37,38,39].

The results of the work show that the presence of water and the exposure of STERED to freezing and thawing do not have a significant effect on the mechanical properties of this material. None of the evaluated groups showed significant variations in the modulus of elasticity values, with the majority of the samples having almost identical stresses. However, some samples had slightly different results due to the inhomogeneity of the material representation of the different components of which STERED is composed. Further experimental measurements are required to refine the results and to clarify the effect of a larger temperature range or number of cycles on its mechanical properties [40,41,42,43].

The use and production of this material has partly contributed to the solution of one of the many problems of our time: the question of “What to do with the produced waste?”. Therefore, if we can find other ways to use this material, we will not only help the automotive industry get rid of this waste but also the entire planet, thereby improving the quality of life for each and every one of us [44,45,46,47].

## Figures and Tables

**Figure 1 polymers-16-00663-f001:**
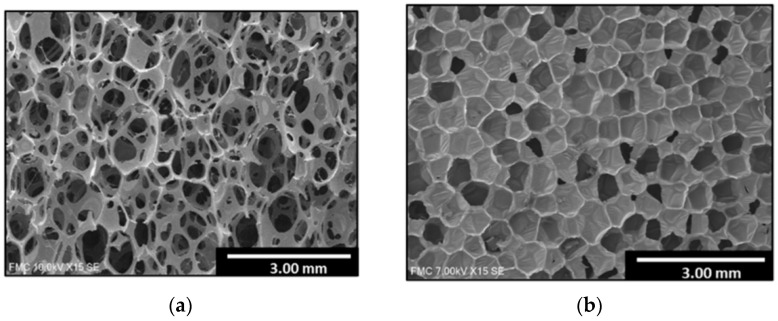
(**a**) Open cell structure. (**b**) Closed cell structure [18].

**Figure 2 polymers-16-00663-f002:**
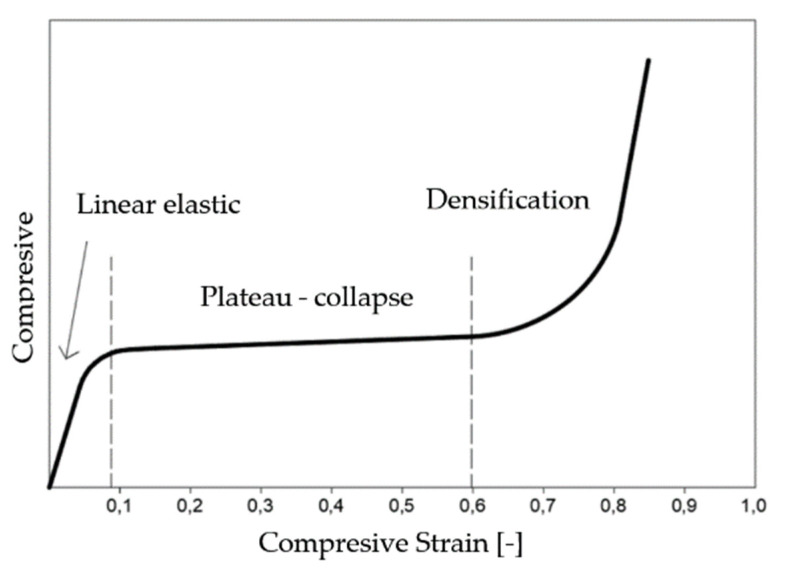
Typical stress–strain curve for polyurethane foams in compression [21].

**Figure 3 polymers-16-00663-f003:**
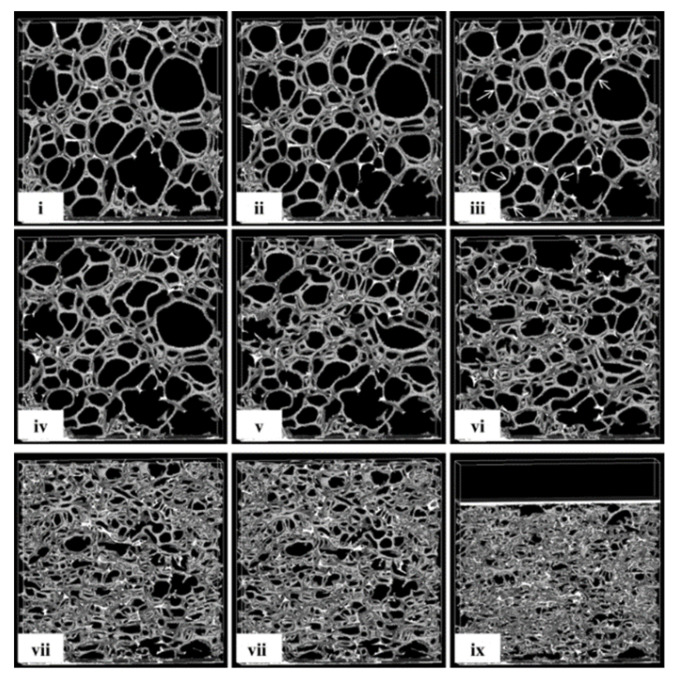
Selected X-ray tomographic images of deformed PU foams as function of pressure [23]. Reprinted with permission from Elliott et al. (2002). 2002 Springer Nature. (**i**–**ix**) stages of change in material cell structure as the stress on the specimen gradually increases.

**Figure 4 polymers-16-00663-f004:**
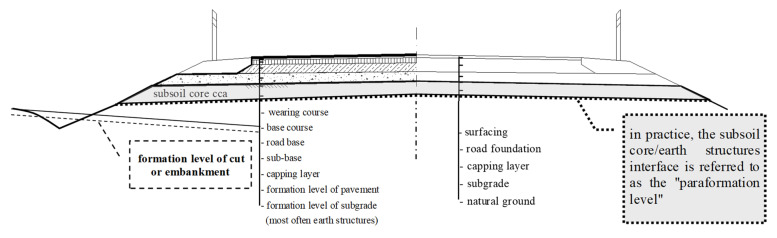
Nomenclature of asphalt roadway layers.

**Figure 5 polymers-16-00663-f005:**
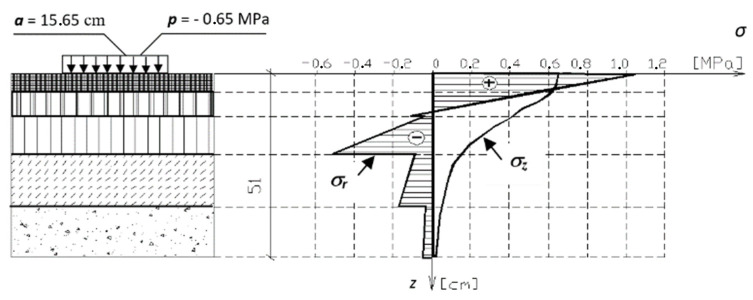
Process of vertical stresses *σ_z_* and radial stresses *σ_r_* in the asphalt roadway structure from loading with a circular surface with radius *a* = 15.65 cm and contact pressure *p* = 0.65 MPa.

**Figure 6 polymers-16-00663-f006:**
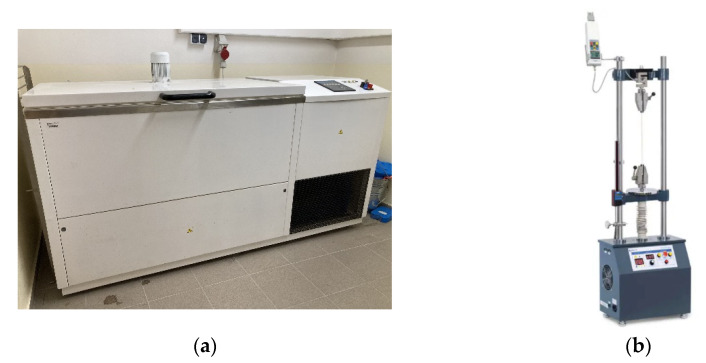
(**a**) Freeze–thawing chamber KD-20-T3.1 [19]. (**b**) Sauter TVM-N test stand.

**Figure 7 polymers-16-00663-f007:**
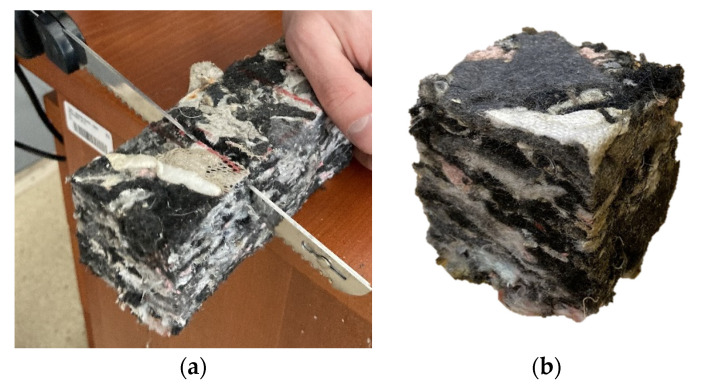
(**a**) Preparation of experimental sample. (**b**) Prepared small scale sample.

**Figure 8 polymers-16-00663-f008:**
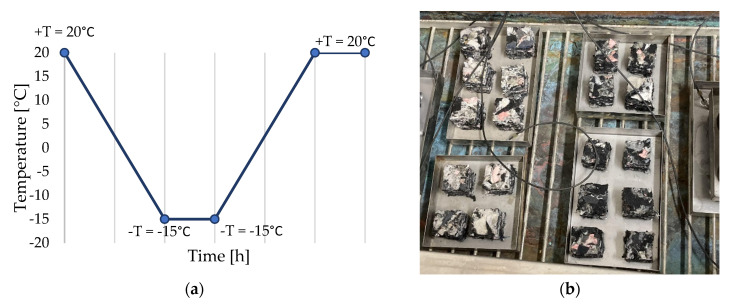
(**a**) One cycle of freeze–thawing of samples. (**b**) Samples laid out in freezing chamber.

**Figure 9 polymers-16-00663-f009:**
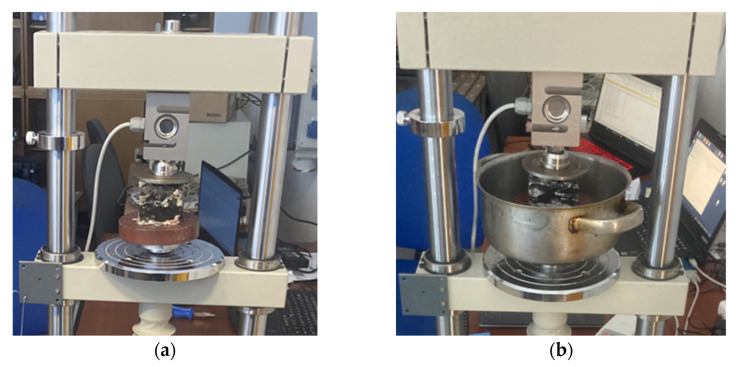
(**a**) Untreated sample. (**b**) Saturated sample.

**Figure 10 polymers-16-00663-f010:**
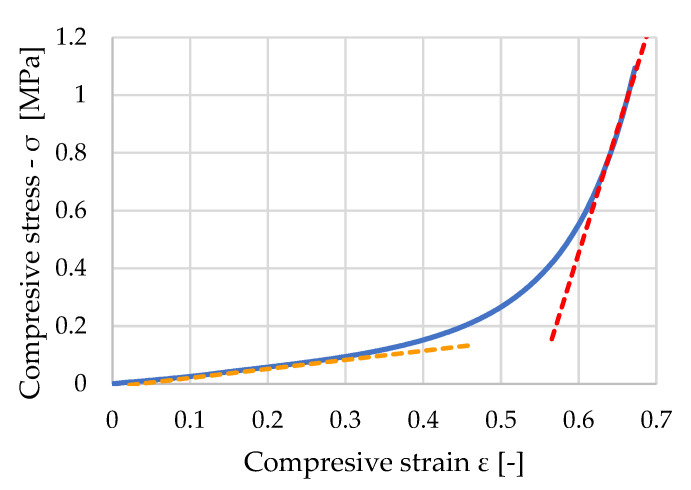
Stress–strain diagram of untreated sample n. 230214-0-05.

**Figure 11 polymers-16-00663-f011:**
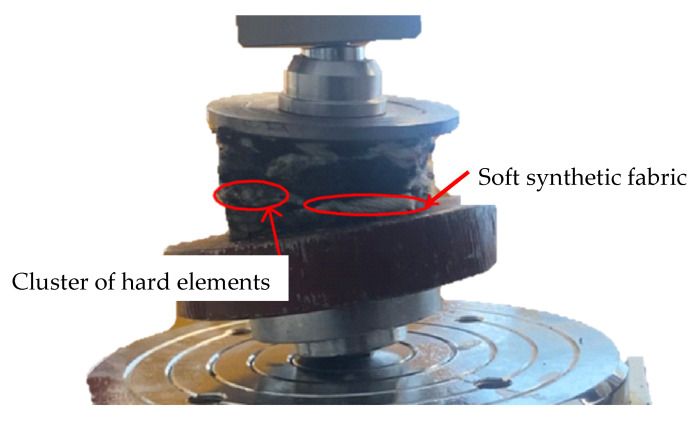
Short description of inhomogeneities of sample.

**Figure 12 polymers-16-00663-f012:**
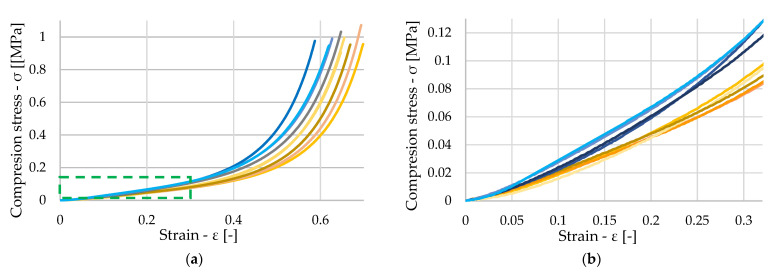
(**a**) Stress–strain diagram of untreated samples (**b**) Detailed view of small deformation.

**Figure 13 polymers-16-00663-f013:**
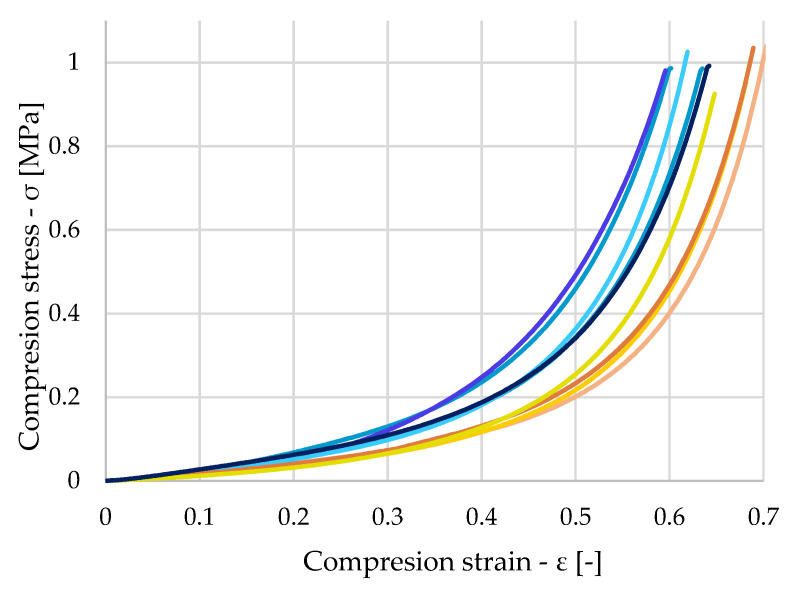
Stress–strain diagram of saturated samples.

**Figure 14 polymers-16-00663-f014:**
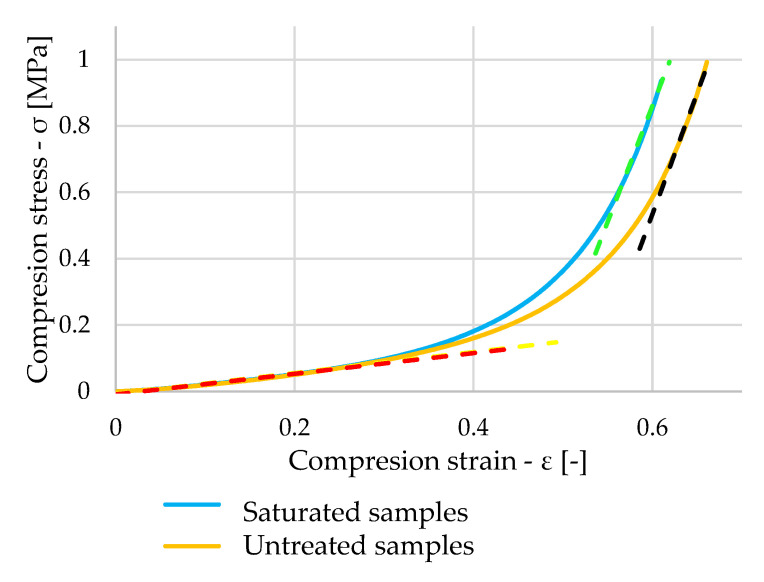
Comparison of stress–strain diagram of untreated and saturated samples.

**Figure 15 polymers-16-00663-f015:**
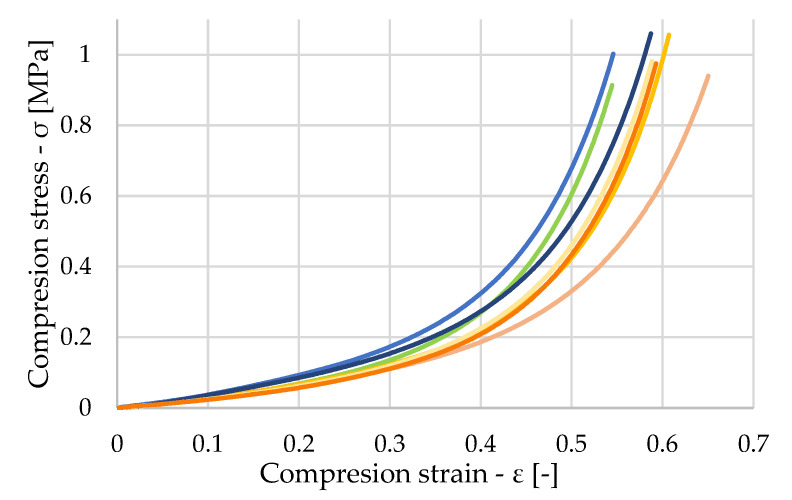
Stress–strain diagram of untreated freeze–thawed samples.

**Figure 16 polymers-16-00663-f016:**
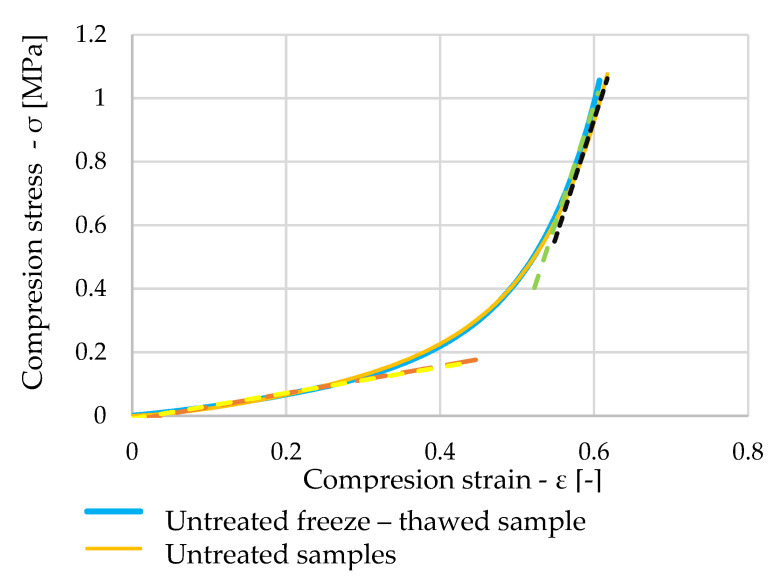
Comparison of stress–strain diagram of untreated and untreated freeze–thawed samples.

**Figure 17 polymers-16-00663-f017:**
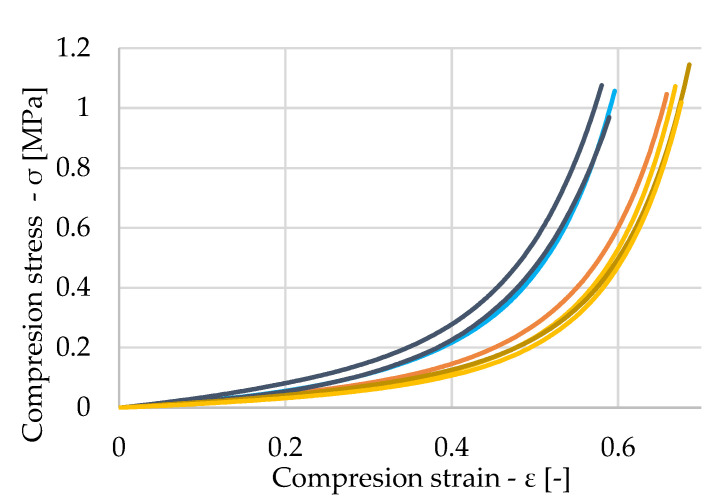
Stress–strain diagram of saturated freeze–thawed samples.

**Figure 18 polymers-16-00663-f018:**
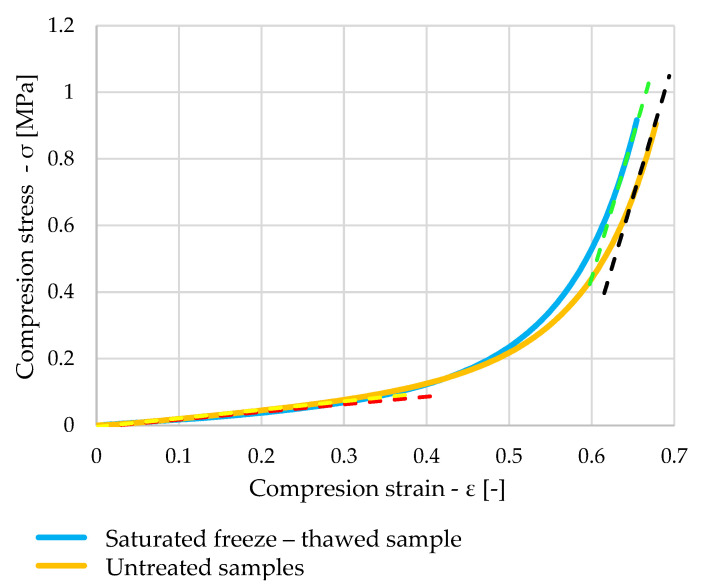
Comparison of stress–strain diagram of untreated and saturated freeze–thawed samples.

**Figure 19 polymers-16-00663-f019:**
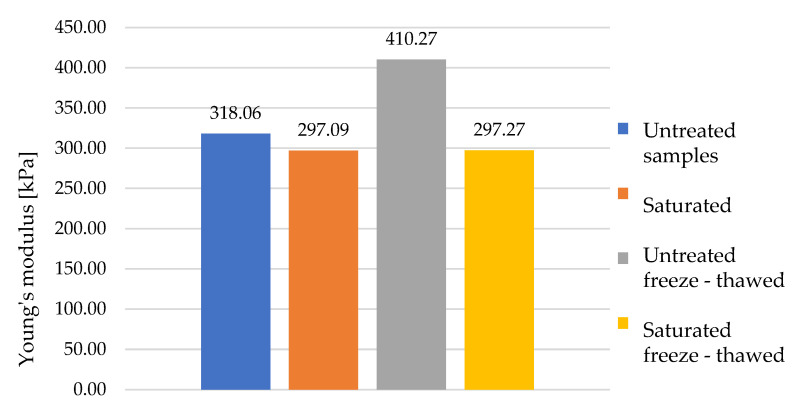
Comparison of Young’s modulus median of all sample groups at plateau.

**Figure 20 polymers-16-00663-f020:**
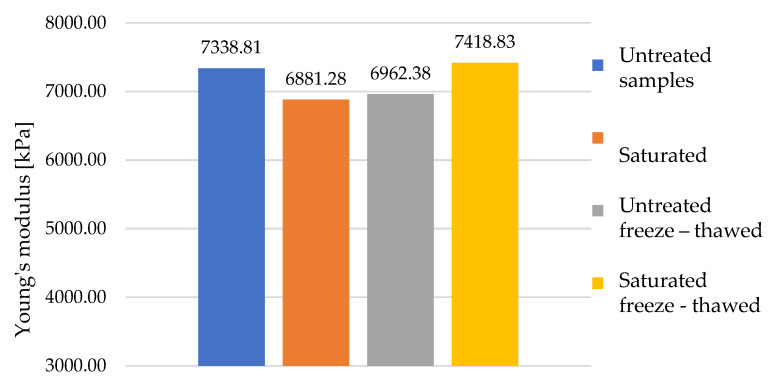
Comparison of Young’s modulus median of all sample groups at densification.

**Table 1 polymers-16-00663-t001:** Selection of samples from each group and its dimensions.

Sample Group	Label of Sample	Height [mm]	Dimensions Width/Length/Height [mm]
0.—untreated	230214-0-01	55.8/57.3/57.6/57.5	48.1/47.3/57.05
	230214-0-02	55.8/55.0/56.1/56.3	51.5/54.9/55.80
	230214-0-03	50.5/47.9/48.8/50.9	54.6/52.1/49.53
1.—fully saturated	230228-1-01	60.0/58.4/58.6/58.5	51.4/49.8/58.88
	230228-1-01	56.4/55.1/55.9/52.3	49.2/49.7/54.93
	230228-1-03	58.4/56.2/57.1/54.9	51.2/50.3/56.65
2.—freeze–thawed untreated	230413-2-01	56.8/56.2/55.9/56.4	48.7/50.5/56.33
	230413-2-02	56.8/57.4/56.2/57.0	53.7/51.2/56.85
	230413-2-03	58.7/58.2/58.2/57.8	52.1/49.6/58.23
3.—freeze–thawed fully saturated	230413-3-01	50.6/51.8/51.4/53.4	49.8/52.3/51.80
	230413-3-02	50.2/50.4/50.9/50.7	47.8/49.1/50.55
	230413-3-03	49.7/50.4/49.9/50.6	47.5/49.8/50.15

**Table 2 polymers-16-00663-t002:** Values of Young’s modulus of untreated samples.

Sample Label	Young’s Modulus—Plateau Collapse [kPa]	Youngs Modulus—Densification [kPa]
0_07	258.8	7539.3
0_10	293.9	7164.5
0_14	257	8275.7
0_16	284.9	7743.9
0_18	268.2	7009.1
0_05	372.3	8766.4
0_12	360.3	8020.8
0_17	382.9	7463.2
0_19	388.1	6376.4

**Table 3 polymers-16-00663-t003:** Weight change in samples before and after saturation.

Sample Label	Weight before	Weight after	Weight after
	Saturation [g]	Saturation [g]	Compression [g]
1_01	34	154	86
1_02	31	131	59
1_03	34	149	79
1_04	25	146	73
1_05	30	139	72
1_06	28	132	66
1_08	29	129	63
1_09	30	148	77
1_10	26	141	78

**Table 4 polymers-16-00663-t004:** Values of Young’s modulus of saturated samples.

Sample Label	Young’s Modulus—Plateau Collapse [kPa]	Youngs Modulus—Densification [kPa]
1_02	241	8114.2
1_04	213.4	7099.5
1_08	228.6	6966.9
1_09	209.1	6953.8
1_01	430.4	6451.2
1_03	331.5	5971.5
1_05	343.5	6458
1_06	316.8	6987.1
1_10	359.6	6929.4

**Table 5 polymers-16-00663-t005:** Values of Young’s modulus of untreated freeze–thawed samples.

Sample Label	Young’s Modulus—Plateau Collapse [kPa]	Youngs Modulus—Densification [kPa]
2_04	405.4	7417.5
2_05	367.8	5870.1
2_09	385.2	6524.8
2_02	404.1	6943.6
2_01	545.1	7099.3
2_07	510.3	7131.5

**Table 6 polymers-16-00663-t006:** Weight change in saturated samples before and after freeze–thawing cycles.

Sample Label	Weight before Saturation [g]	Weight before Freeze–Thaw Cycles [g]	Weight after Freeze–Thaw Cycles [g]
3_01	31	138	86
3_02	28	144	69
3_03	27	139	61
3_04	34	127	79
3_05	30	145	63
3_06	34	129	38
3_07	31	146	66
3_08	34	130	63
3_09	25	141	75
3_10	30	152	80

**Table 7 polymers-16-00663-t007:** Values of Young’s modulus of saturated freeze–thawed samples.

Sample Label	Young’s Modulus—Plateau Collapse [kPa]	Youngs Modulus—Densification [kPa]
3_04	265.2	7357.3
3_06	229.8	8462
3_09	238.3	8131.4
3_10	192.7	7098.9
3_01	346.2	7506.2
3_07	346.6	6660.1
3_08	462.1	7759

## Data Availability

The data presented in this study are available on request from the corresponding author. At the time the project was carried out, there was no obligation to make the data publicly available.

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
