# Peer review of "Identification of Hybrid Polymer Material STERED and Basic Material Properties Used in Road Substructures or Pavements"

_polymers, 2024, doi:10.3390/polym16050663_

Round 1

Reviewer 1 Report

Comments and Suggestions for Authors

The manuscript contains innovative application of STERED but some more investigations are needed.

1. The samples have been evaluated with respect to compressive test. what is the justification behind this? On the road there is also lot of shearing and hardness is also an important factor.

2. The morphology of the samples should be tested for proper analysis of ununiform strength characteristics of the samples. 

Author Response

Dear reviewer, please find the response to the review in the attached document.

Reviewer 2 Report

Comments and Suggestions for Authors

Overall comments: The following research questions are not clarified in this research and should be answered to improve the quality of paper.

(1) Is it enough to demonstrate the mechanical properties of STERED material by only freeze-thaw testing?

(2) Compared to the conventional subbase materials for road (e.g., gravels or reclaimed asphalt pavement), are the mechanical properties of STERED better or comparable?

(3) Is there risk of leaching if STERED is applied as subgrade?

Detailed comments:

1. Introduction:

(1) References are need for the sentence “The changing climatic conditions of Central Europe, a decrease total rainfall and a significant increase in torrential rainfall are facts”.

(2) The paragraph “This material is currently being used in sound-absorbing…” is still the content of previous paragraph. It is not reasonable to put them in a separate paragraph.

(3) References are need for the sentence “Based on an indicative quantification of elastic moduli of 5 to 10 MPa ???, it was found that PROTERED (STERED) should be placed in unbound road subbases”. Besides, delete the question marks in the sentence.

(4) The paragraph between line 41-49 does not provide convincing reasons why STERED material should be used in road construction (compared to their use in sound-absorption). It is more suitable to put the contents of Section 1.4 in this paragraph and remove Section 1.4. Moreover, authors should also explain why only compression testing is focused without considering other mechanical properties of asphalt.

(5) It is necessary to mention that STERED material is a type of polymeric foam before Section 1.1. Otherwise, it is strange to introduce polymeric foam at the beginning.

(6) References are need for the sentence “The use of polymeric foams in engineering applications is steadily increasing, mainly due to their energy absorbing ability, low weight and relatively low production costs. The main applications of soft and semi-rigid foams include insulation, packaging and cushioning, while rigid foams are also used in structural applications as well as the core of lightweight sandwich structures”.

(7) Figure 1 should be mentioned in the main text of Section 1.1 (instead of putting the figure without mentioning it).

(8) References are need for the sentence “Polymeric foam materials may exhibit local variability, which occurs due to different manufacturing methods and different structure of the material, resulting in different mechanical properties”.

(9) Use the same citation format for Figure 2 as that for the main text (instead of using DOI link).

(10) “Figure 1.3” in line 90 should be “Figure 3”.

(11) References are need for the sentence “Previous research has shown that the initial linear elastic portion of the stress-strain curve is the result of bending of the cell walls”. What are the “previous research”?

2. Methodology

(1) References are need for the sentence “The SAUTER TVM 30KN70N test stand was used for destructive experimental measurement of compression strain”. The full information of the testing standard should be available in the reference list.

(2) There is lack of units in Table 1.

(3) Typo “previouse capitol”.

3. Results

(1) Figure 10 is still the part of methodology (data processing). The figure along with the corresponding main text should be put in Section 2.

(2) Section 3.1 should also be put in Section 2 (they are not final results).

(3) Typo “Figure 12F”.

(4) There is lack of legend in Figure 12, 13, 15, and 17. What do different color represent in these figures?

4. Discussion

(1) There is no “green column” in Figure 19. Why do authors mention green column in the main text? Same for “red and purple columns”.

Comments on the Quality of English Language

Several typos and inappropriate paragraph separations were found.

Author Response

(The authors gave the same response as above.)

Round 2

Reviewer 2 Report

Comments and Suggestions for Authors

Most comments have been addressed. There are some minor comments based on previous review:

1. Introduction

(1) Old comment: References are need for the sentence “Based on an indicative quantification of elastic moduli of 5 to 10 MPa ???, it was found that PROTERED (STERED) should be placed in unbound road subbases”. Besides, delete the question marks in the sentence.

New comment: It is good that the authors corrected the typo. However, references are still not added. Where does “5 to 10 MPa” come from?

3. Results

(1) Old comment: There is lack of legend in Figure 12, 13, 15, and 17. What do different color represent in these figures?

New comment: It is good that the authors added legends, but there seems to be display problems. The legends of different colors in Figures 14, 16, and 18 are overlapped with each other. This should be improved to show a clear legend.

Comments on the Quality of English Language

Major typos have been corrected.

Author Response

Many thanks to the reviewer for double-checking and substantive comments.
